# Integrating Visual Cues via Prompting for Low-Resource Multimodal Named Entity Recognition

## Abstract

In the field of Natural Language Processing (NLP), the task of Named Entity Recognition (NER) is quite established. However, most existing methods predominantly rely on textual data alone, overlooking the information that can be derived from other modalities such as images. This issue is particularly pronounced in low-resource settings, where the absence of extensive labeled data can significantly impede the performance of NER systems. Existing solutions, while attempting to address this limitation, often require comprehensive fine-tuning and are not readily applicable in such low-resource conditions. This research confronts these challenges by proposing a novel approach to Multimodal Named Entity Recognition (MNER) under low-resource constraints. We recast the MNER task as an open-ended question-answering problem, particularly suitable for modern generative language models. Our findings provide novel insights into the complex interplay between model design, prompt crafting, and training data characteristics that determine the efficacy of visual integration. The strengths and limitations elucidated can inform future efforts at the intersection of multimodal representation learning, generative modeling, and prompting.

## 1 Introduction

**Context.** Multimodal learning, combining signals from diverse modalities like vision and language, offers promising opportunities to enhance natural language processing (NLP) systems. Named entity recognition (NER), focused on detecting and classifying textual entity mentions, is a prime candidate task for such multimodal augmentation. In fact, multimodal NER (MNER) has been an active area of research since 2018 Lu et al. (2018); Moon et al. (2018); Zhang et al. (2018); Chen et al. (2021); Sun et al. (2021b); Zhao et al. (2022). Recent advances in large-scale generative pre-trained language models provide new potential to re-examine multimodal NER (MNER), particularly under low-resource constraints.

Large generative models, such as GPT-3 and GPT-4, have been used in various tasks, including few-shot learning Brown et al. (2020); OpenAI (2023); Teehan et al. (2022). These models have remarkable capabilities in understanding and generating human-like text based on the prompts provided to them.

**Challenge.** Despite advancements, NER faces several challenges, including the scarcity of resources and the morphologically rich behavior of languages in part because of the lack of annotated data and the absence of domain-specific resources Vijayvergia et al. (2022). There is little prior research that considers the details of few-shot learning using prompting for multimodal datasets. As a result of this knowledge gap, we study few-shot learning using prompting for multimodal datasets in an empirical fashion in this paper.

**Contribution.** This work specifically investigates the efficacy of prompt-based conditioning as a technique to effectively incorporate visual information from images to improve MNER using modern generative models. Our approach relies on reformulating the standard MNER classification task as open-ended question answering, conditional on model inputs comprising textual content, relevant images, and crucially, template-based instructions and demonstrations.

Our core contributions include:

- Demonstrating prompting multimodal generative models for few shot named entity recognition

- Show significant sensitivity of how a visual modality may be impacted by the prompt posed

- Find promising results for zero-shot learning on publicly available multimodal datasets

- Clarifying the intricate relationship between visual improvements and the features and representations of training data

## 2 RELATED WORK

This section explores previous work in Named Entity Recognition (NER) and Multimodal Named Entity Recognition (MNER), with a focus on the use of prompts.

### 2.1 NAMED ENTITY RECOGNITION (NER) AND PROMPTING

NER is a subfield that focuses on identifying and classifying entities within text into predefined categories. The use of few-shot learning in NER has been a notable advancement, allowing for the development of effective models in settings with limited resources Cariello et al. (2021). The low-resource environment can be attractive due to the lack of available labeled data and the expense of creating such Perez et al. (2021); Liu et al. (2022). The most low-resource environment is then one where there is no dev-set, named true few-shot learning by Perez et al. (2021). Prompting-based NER methods have gained substantial attention for effectively recognizing named entities using prompts or questions to guide the NER system Rozhkov & Loukachevitch (2023). A question-answering approach was done by Liu et al. (2022) with reportedly good results.

### 2.2 MULTIMODAL NAMED ENTITY RECOGNITION (MNER)

MNER is an approach in NER that uses information from different sources, such as text and images, to improve the accuracy of entity recognition tasks Moon et al. (2018); Zhang et al. (2018). Despite potential issues such as noise and conflicting information, MNER has been used to enhance entity recognition tasks Chen et al. (2021).

Apart from prompting, other techniques have been explored in the realm of MNER. These methods include the extraction of text from images for additional contextual information Wang et al. (2021), the use of forget gates for selecting visually relevant information Sun et al. (2021a), and the employment of graph-based techniques for multimodal alignment Zhang et al. (2021).

#### 2.2.1 PROMPT-BASED METHODS FOR MNER

Prompt-based methods have been adapted for MNER. PromptMNER uses similarity scoring between image-aligned prompts and inputs for classification Wang et al. (2022a). This is done using premade prompts corresponding to each entity type to gain a similarity score for enhanced classification information.

In conclusion, while significant progress has been made in the field of NER and MNER, several challenges remain. Continued research and development of more advanced models and approaches are essential to further enhance the performance and effectiveness of NER and MNER systems.

## 3 METHODS

This section explores transforming multi-class named entity recognition (MNER) into an open-ended question-answering task. We borrow heavily from Liu et al. (2022) in the framing of these prompts, mainly because it was conceptually simple and could be used in zero-shot prompting. This transformation involves constructing prompts, represented as $P$, with various elements including an entity type $E$, a text segment $T$, and the model's output $A$.

### 3.1 PROMPT DESIGN AND STRUCTURE

The design of the prompt $P$ is simple and functional, including the following elements:

- **Instruction** ($I$)**:** Instructions provide basic guidelines. For example, "We want to do entity recognition. If there is no entity the answer is 'none'. If there are multiple answers, output them with '&' between."
- **Question** ($Q$)**:** The question part, such as "What in the text corresponds to a [ENT] entity?", is structured for clarity.
- **Text** ($T$)**:** Contains the text segment for NER analysis.
- **Answer** ($A$)**:** Holds the model's output.
- **Training Examples** ($TE$)**:** Contextual examples in the form Question: [Q] Text: [T] Answer: [A] guide the model. In this experiment we report zero-shot results without using these in-context training examples.

For querying a specific entity type $E$, a consistent structure is used. For example:

- **Query for Person:** Use `PROMPT[person]` with `TEXT_N` to obtain `ANSWER[person]_N`.

This gives specificity to the query on one hand, but on the other, introduces inference run complexity by a factor of the number of entity types, $E$. Each entity type requires a unique prompt, leading to an increase in the total number of prompts that must be run:

$$\text{Complexity} \propto E.$$

### 3.1.1 INSTRUCTIONS

Instructions are helpful for effective prompting (Mishra et al., 2022). A set of instructions, covering various scenarios including image specification, has been established. Training examples are selected from the dataset, ensuring a balance in entity types and adherence to model constraints. The instructions are as follows:

The instruction text is formed as:

> We want to do entity recognition. The image is meant to help with the question. If there is no entity the answer is none. If there are multiple answers, output them with '&' between.

### 3.2 EXTRACTING ANSWERS AND RESOLVING CONFLICTS

In extracting an answer, denoted as $A$, from a model, ensuring that the answer aligns with the source text $T$ is essential. An answer $A$ is considered valid only if contained within $T$; otherwise, it is marked invalid. This validity check is represented by:

$$A = \begin{cases} A & \text{if } A \subseteq T \\ \text{invalid} & \text{otherwise.} \end{cases}$$

When faced with conflicting answers, the most reliable one, denoted as $A^*$, is determined based on the highest average confidence score. This score is calculated from the output logits, which are measures of the model's certainty for each possible answer. Let $\mathcal{A}$ be the set of all potential answers. For each answer $A$, its confidence score is denoted by $c(A)$. The answer with the highest average confidence score $A^*$ is then defined as:

$$A^* = \arg\max_{A \in \mathcal{A}} \bar{c}(A),$$

where

$$\bar{c}(A) = \frac{1}{|A|} \sum_{i=1}^{|A|} c(A_i).$$

In this formula, $c(A_i)$ represents the confidence score of the $i$-th element or aspect of answer $A$, and $|A|$ is the count of these elements or aspects within the answer $A$.

### 3.2.1 QUESTION PROMPT TEMPLATES

Various question templates are employed to identify entities in the text. Inspiration here is also taken from Liu et al. (2022), but with what is mostly a crude attempt at constructing simple questions with subjectively minor variations. Below are the question prompt templates used and their corresponding numbering:

1. What word(s) in the text corresponds to a [ENT] entity?
2. [ENT] in text?
3. [ENT] entity in text?
4. What is the [ENT] entity in the text?
5. What is the [ENT] in the text?

## 4 EXPERIMENT AND RESULTS

This section presents the experimental setup and the obtained results. The experiments were conducted on two publicly available Twitter datasets, Twitter2015 and Twitter2017, which consist of text-image pairs. The models' performance was evaluated using mean micro F1 scores across question prompt templates.

### 4.1 DATASETS

Table 1: Experimental datasets for Twitter-2015 and Twitter-2017.

| Category | Twitter-2015 | | | Twitter-2017 | | |
|---|---|---|---|---|---|---|
| | **Training** | **Development** | **Testing** | **Training** | **Development** | **Testing** |
| Person | 2217 | 552 | 1816 | 2943 | 626 | 621 |
| Location | 2091 | 522 | 1697 | 731 | 173 | 178 |
| Organization | 928 | 247 | 839 | 1674 | 375 | 395 |
| Miscellaneous | 940 | 225 | 726 | 701 | 150 | 157 |
| Total | 6176 | 1546 | 5078 | 6049 | 1324 | 1351 |
| Number of Tweets | 4000 | 1000 | 3257 | 3373 | 723 | 723 |

We evaluate our proposed multimodal named entity recognition (MNER) approach on two publicly available Twitter datasets containing text-image pairs: Twitter2015 (Zhang et al., 2018) and Twitter2017 (Lu et al., 2018). Statistics on the datasets used are provided in Table 1. These datasets encompass tweets spanning diverse topical domains including news, entertainment, and daily experiences. Both datasets annotate entities with the four types used in the CoNLL-2003 shared task: persons, organizations, locations, and miscellaneous. Since our reported experiment is only zero-shot, we utilize only the testing dataset.

Notably, these are currently two of the few openly accessible multimodal datasets for named entity recognition. While Moon et al. (2018) also constructed a proprietary MNER dataset from Snapchat, its private status precludes external evaluation and comparison. The limited public availability of large-scale, multimodal NER corpora represents a key challenge for advancing this research area. We discuss avenues to address this limitation and facilitate further progress in subsection 5.4.

A summary of the computing resources used for experiments is provided in Table 2.

### 4.2 EVALUATION METRICS

The following evaluation uses a micro F1 score using a stringent NER metric, ensuring that predictions are correct only if they match all tokens and the right entity type.

We are interested not in determining a single best prompt, but finding the mean performance and the variability and as such we average the F1 scores over results from the five prompts to gain an understanding of the overall performance and the variability of similar prompts.

Table 2: Hardware Specifications of the Experimental Computing Cluster.

| Hardware Specification | Details |
| --- | --- |
| Node Type | Dell DSS8440 |
| Number of Nodes | 2 |
| CPUs per Node | 2 x Intel Xeon Gold 6248R |
| CPU Cores per Node | 48 |
| RAM (GB) | 80 |
| GPU | A100 80GB |

When attempting to compare the lack of information from the image in the dataset, we introduce a counterfactual image whereby it should have no information, therefore using a black image, denoted later as using no image.

## 4.3 MODELS

Table 3: Relevant Inference Parameters for experiment.

| Parameter | Comment |
| --- | --- |
| Temperature | Set to 0 for negation of randomness. |
| Inference Batch Size | Some models acted unstable with batches higher than 1, so batch inference was run at 1. |
| Search Parameter | Greedy decoding/Greedy search. |
| Max New Tokens | Set to 20 to limit the length of answers that were likely to be generated out of the models. |
| Max Input Length | Models were configured to 1024 tokens without truncation. |

We evaluate the few-shot learning performance of several state-of-the-art multimodal models on our tasks that have shown strong capabilities using few-shot instructions, including FLAN-T5-XL, FLAN-T5-XXL, BLIP2-FLAN-T5-XL, BLIP2-FLAN-T5-XXL, InstructBLIP-FLAN-T5-XL and InstructBLIP-FLAN-T5-XXL Chung et al. (2022); Li et al. (2023); Dai et al. (2023), with relevant inference parameters in Table 3.

**FLAN-T5** was released by Chung et al. (2022) and is instruction fine-tuned on many NLP tasks, also named entity recognition. They propose improved scaling protocols that achieve similar fine-tuning quality with 50% fewer parameters and 40% faster training compared to T5-base. We use the XL and XXL-sized models, which are 3B and 11B parameters respectively.

**BLIP-2** is a method for transferring the few-shot learning ability of auto-regressive language models to a multimodal setting Li et al. (2023). By training a vision encoder to represent images as continuous embeddings, in this case, CLIP Radford et al. (2021), and using a pre-trained language model, we use the FLAN-T5 version, BLIP-2 can learn new tasks with just a few examples. It's been reported to rapidly learn words for new objects, perform visual question-answering with limited examples, and use outside knowledge. BLIP-2 achieves superior or competitive performance compared to similar methods while training fewer parameters.

**InstructBLIP** is an model that while using the BLIP2 architecture is specifically tailored for and fine-tined on instructive tasks Dai et al. (2023). So differing from BLIP2 it's instruction-tuned in a multimodal setting. Also here CLIP and FLAN-T5 is used.

These multimodal models demonstrate promising learning abilities from limited labeled examples by leveraging large pre-trained language models alongside CLIP's strong vision capabilities. Their multimodal fusion strategies aim to ground language in the visual context effectively.

Table 4: Micro F1 Scores and Standard Deviations for the Twitter2015 Dataset: This table summarizes mean scores for two scenarios: with images and without images. The FLAN T5 model is unimodal and does not have image scenarios. Scores are represented in percentage.

| Model | Size | With Image | Zero-shot F1 (%) |
|---|---|---|---|
| InstructBLIP-Flan T5 | XL | ✓ | $48.19 \pm 6.98$ |
| | XL | × | $51.35 \pm 2.49$ |
| | XXL | ✓ | $54.27 \pm 5.22$ |
| | XXL | × | $53.31 \pm 6.80$ |
| BLIP2-Flan T5 | XL | ✓ | $58.07 \pm 5.66$ |
| | XL | × | $57.60 \pm 4.43$ |
| | XXL | ✓ | $50.38 \pm 7.07$ |
| | XXL | × | $49.18 \pm 8.89$ |
| FLAN T5 | XL | - | $59.67 \pm 3.48$ |
| | XXL | - | $48.37 \pm 5.99$ |

Table 5: Micro F1 Scores and Standard Deviations for the Twitter2017 Dataset: This table summarizes mean scores for two scenarios: with images and without images. The FLAN T5 model is unimodal and does not have image scenarios. Scores are represented in percentage.

| Model | Size | With Image | Zero-shot F1 (%) |
|---|---|---|---|
| InstructBLIP-Flan T5 | XL | ✓ | $40.28 \pm 8.30$ |
| | XL | × | $40.76 \pm 9.44$ |
| | XXL | ✓ | $48.37 \pm 4.69$ |
| | XXL | × | $51.04 \pm 5.42$ |
| BLIP2-Flan T5 | XL | ✓ | $51.02 \pm 6.64$ |
| | XL | × | $52.11 \pm 6.77$ |
| | XXL | ✓ | $49.47 \pm 6.07$ |
| | XXL | × | $52.03 \pm 6.55$ |
| FLAN T5 | XL | - | $56.68 \pm 2.91$ |
| | XXL | - | $50.01 \pm 5.76$ |

## 4.4 RESULTS

Results in Tables 4 and 5 showcase model performance with and without images for the two datasets. Figure 1 and Figure 2 provide F1-scores for each entity type, for the Twitter2015 and Twitter2017 dataset respectively.

## 5 ANALYSIS

This section provides an analysis of the results, focusing on the multimodal aspects of Named Entity Recognition (NER) and discussing implications for future research.

### 5.1 VALIDITY OF RESULTS

The evaluations were conducted using established datasets and metrics to facilitate reproducible comparison. However, limitations exist. The Twitter datasets used represent a narrow range of domains and entity types. The evaluation primarily focused on exact string matching, which does not account for semantic equivalence. Future research should consider exploring a more diverse range of data and evaluation methods to enhance the comprehensiveness of the analysis.

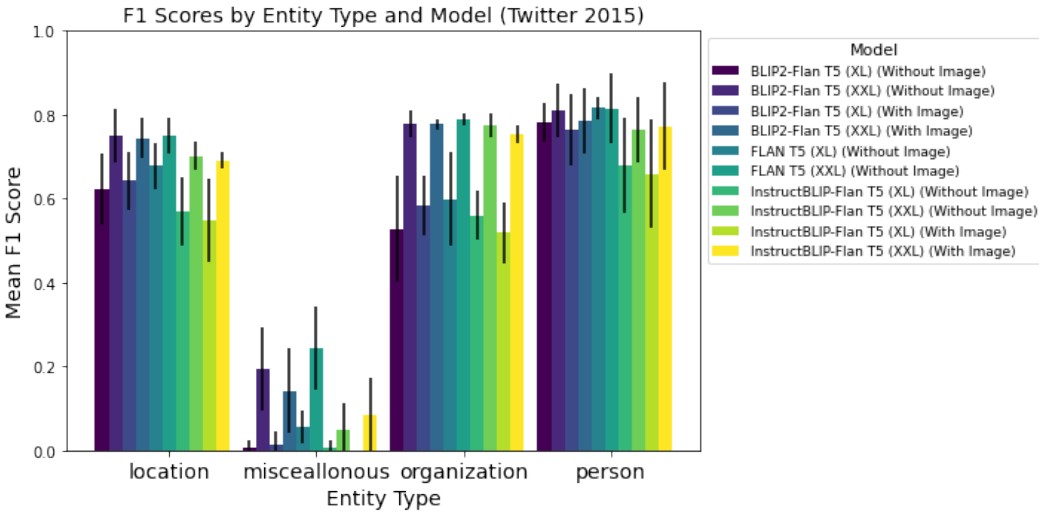

Figure 1: Mean F1 Scores by Entity Type and Model for the Twitter 2015 dataset. The plot shows the mean F1 scores with standard deviation error bars for different models with and without images.

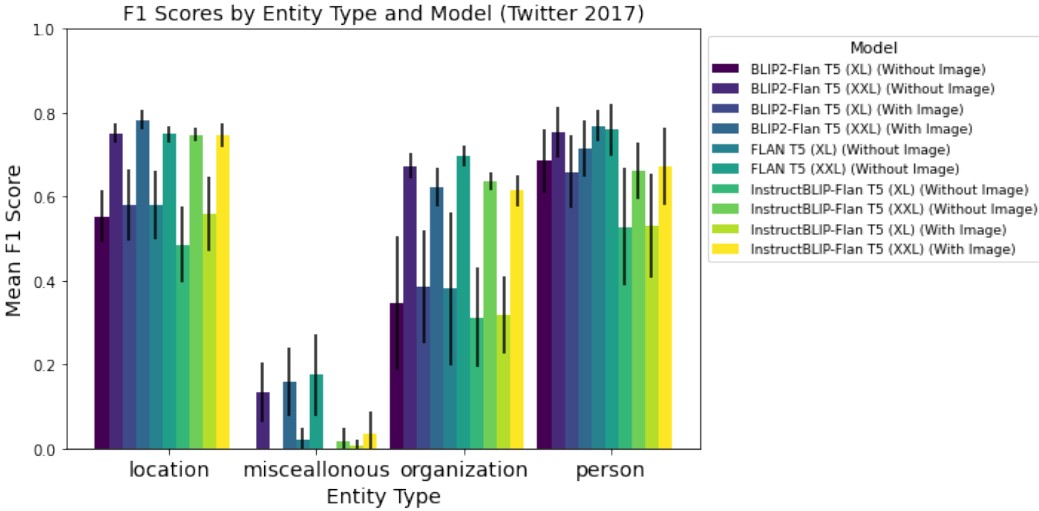

Figure 2: Mean F1 Scores by Entity Type and Model for the Twitter 2017 dataset. The plot shows the mean F1 scores with standard deviation error bars for different models with and without images.

In the context of prompt selection, we aimed not merely to identify a single best prompt but rather to evaluate the average performance and variability across multiple prompts. This approach allowed us to gain insight into the overall performance and the variability of similar prompts. We noticed that the survivorship bias, which usually arises when worst-performing prompts are overlooked, could potentially affect the results. Therefore, we also included crude prompts in our experiment to examine their overall mean performance and volatility.

However, using pre-trained models, which also make use of web-corpora and have been instruction-tuned on named entity recognition, is a confounding factor for pre-trained LLMs in general, but specifically in our case. This was only alleviated because the relevant pre-training regarding the dataset happened mainly in the unimodal case for the FLAN-T5 model.

## 5.2 PERFORMANCE

Analyzing the general performance of our models in Table 4 and Table 5, we observe a relatively high performance considering the low-resource setting of our task. Although the performance does not quite reach the level of fully trained fine-tuned models, surveyed results provided in Table 6 it is still notable given the resource constraints (zero shot). For instance, we achieved around 50%+ F1 scores. This is promising compared to the state-of-the-art results, which usually range around 70-90%. The lack of benchmarks for low-resource MNER poses a challenge for performance evaluation relative to other research.

Our models performed well on specific entity types, such as person, location, and organization, as seen in Figure 1 and Figure 2. However, the performance on entities like 'miscellaneous' was not as high. This is likely due to the lack of contextual information available for the models to discern what entities should be classified as 'miscellaneous'. Thus, the overall score was impacted by the models' lower performance on these types of entities. Evaluating the results may indicate that the models may perform significantly better on more intuitive or well-defined entities.

## 5.3 INTEGRATING VISUAL CONTEXT

Integrating visual context into NER tasks proved to be more complex than initially anticipated. The effectiveness of visual context integration varied significantly across different models and conditions. While some models demonstrated a noticeable improvement when supplemented with images, others showed negligible or even decreased performance. It's essential to note that the choice of prompts played a significant role in this variation, suggesting the need for careful prompt selection and design in multimodal tasks.

Finally, the image validity, particularly the use of a black image as a counterfactual for using an image, needs further investigation. In our study, a black image served as a baseline, representing a scenario where no image information was available. While this approach provided some insights, the interaction between the textual content and this form of image input remains an open question for future research.

## 5.4 DATASET REPRESENTATIVENESS AND EVALUATION

The efficacy of visual context integration heavily relies on the quality and representativeness of the datasets used. The datasets used in this study may not fully encapsulate the potential of multimodal relationships due to their domain specificity and the lack of diverse visual-textual entity associations. To provide a more robust evaluation of visual integration in NER, future work should focus on creating more comprehensive and diverse datasets.

Additionally, the unbalanced nature of datasets and the metrics paid attention to or weighed for evaluating which results actually show the strongest performance is an open question. The overall F1 metric is not an end-all and be-all of evaluation, and as we show entity-level performance can give nuance to the results, as undoubtedly could more extensive evaluation such as disseminating the precision, recall, overall or entity level, other metrics, partly correct metrics for NER, and other varieties of scoring based on "almost" correct. It is also true that we have investigated by random sampling the corpus and subjectively disagree with the ground truth labels, and followingly seen predictions we believe to be reasonable be scored as false positives.

## 6 CONCLUSION AND FUTURE WORK

We demonstrated low-resource MNER via prompting, with strong zero-shot performance, but also showed that the visual modality in some cases did not improve, but decreased the overall performance using our evaluation methods. Analysis of our results and related work elucidates progress and challenges for this emerging area. Results against state-of-the-art models for MNER are provided in Table 6, which, even though they represent full-resource environments, we believe can be used as a contextual indication given the lack of other low-resource experiments on these datasets.

Future directions include developing more robust and diverse multimodal datasets, evaluation paradigms capturing value of visual cues, and model architectures further enhancing grounding

Table 6: Surveyed results for methods fully-fine-tuned on Twitter2015 and Twitter2017.

| Modality | Method | Twitter2015 | Twitter2017 |
|---|---|---|---|
| Text | BERT Jia et al. (2022a) | 71.32 | 82.95 |
| | BERT-CRF Jia et al. (2022a) | 71.81 | 83.44 |
| Text + Image | ACoA Wang et al. (2022b) | 70.69 | 82.15 |
| | ATTR-MMKG-MNER Jia et al. (2022a) | 73.27 | - |
| | UMT-BERT-CRF Yu et al. (2020) | 73.41 | 85.31 |
| | MAF Xu et al. (2022) | 73.42 | 86.25 |
| | RIVA Sun et al. (2020) | 73.8 | - |
| | MRC-MNER Jia et al. (2022b) | 74.63 | 86.85 |
| | RpBERT Sun et al. (2021a) | 74.8 | 85.51 |
| | UMGF Zhang et al. (2021) | 74.85 | 85.51 |
| | MNER-QG Jia et al. (2022a) | 74.94 | 87.25 |
| | ITA-All+CVA Wang et al. (2021) | 78.03 | 89.75 |
| | PromptMNER Wang et al. (2022b) | 78.6 | 90.27 |

between language and vision. The present release of new multimodal models should facilitate this. Advances at the intersection of these areas hold promise to realize more potential in multimodal learning.

**Code of Ethics**   In conducting this research, we, the authors, adhered to the code of ethics laid out by International Conference on Learning Representations (ICLR).

**Reproducibility Statement**   We have tried to make this work reproducible by including used hardware, see Table 2 and inference parameters see Table 3 in the body of the paper.

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
