# OpenReview forum: "Integrating Visual Cues via Prompting for Low-Resource Multimodal Named Entity Recognition"
_ICLR.cc/2024/Conference — Submitted to ICLR 2024_

### Official Review · Reviewer_Hmr3 · 2023-10-25

**Soundness:** 1 poor
**Presentation:** 2 fair
**Contribution:** 1 poor
**Rating:** 1
**Confidence:** 5

**Summary:**

This work focuses on low-resource multimodal named entity recognition (MNER). The authors reformulate MNER as an open-domain question-answering problem in a low-resource setting and provide analysis.

**Strengths:**

The topic and approach are interesting.

**Weaknesses:**

1. The contribution of this paper is quite limited. The authors use a large number of FLAN-based models (both text-only and multimodal) for multimodal NER tasks.
2. The method is not new. QA for NER can be traced back to [1] and there are recent works using QA for all multimodal IE tasks in both zero-shot and few-shot settings.

[1] Li et al., A Unified MRC Framework for Named Entity Recognition. ACL 2021.

[2] Sun et al., Multimodal Question Answering for Unified Information Extraction. Arxiv 2023.

1. The evaluation is very limited. More models, including both instruction-tuned (IlIava, MiniGPT4, etc.) and base multimodal models (Open Flamingo), should be evaluated. Additionally, analysis of more tasks would be more interesting. While multimodal NER is to-some-extend important, it is not representative enough. Evaluations on other multimodal information extraction tasks would be beneficial.
2. Findings are not striking. With more models evaluated, we may observe more interesting insights and general observations from these models.

**Questions:**

No

---

> ### Author Response · Authors · 2023-11-23
>
> Thank you for your insightful review and the points you have raised regarding our paper. We value this opportunity to address your concerns and further clarify our work.\
> Weaknesses:
> 1. The contribution is indeed limited by the inclusion of few varieties of models. We have done additional experiments, not discussed in the paper due to limited space.  We believe the significance of our MNER study was illustrated well by the experiments and models we have included in the updated version of the paper.
> 2. QA is indeed not new, but the paper mentioned by the reviewer is unimodal, while we focus on the multimodal setting.  Our use of MLLMs specifically to MNER in this prompting fashion on the other hand is. You mention one work [2] published AFTER the ICLR submission deadline, we do not regard it to be fair to use it as an argument against our paper’s contribution. We anticipate that there were numerous recent works in zero and few-shot learning last time. These developments likely span both unimodal and multimodal applications, which may perform similar tasks.
>
>
> 1. We agree with the reviewer that the inclusion of more models would have been a benefit.  Indeed, we have worked on and have results for this, even though we did not publish it here due to limited space. While additional tasks indeed are interesting, this is again a question about depth version breadth for a 9-page ICLR paper.  Our opinion upon submission was that the scope of our paper was similar to that of previously published ICLR papers.
> 2. It’s true that the findings are not confirmatory or positive.  However, negative results should not be and are not banned from the literature if they are deemed to be of interest. Thus, we actually considered the results (that experiments with image modalities did not do better) interesting in themselves. Importantly, the literature on MNER often does not appropriately compare the relative advantage of including the image.  We have added some discussion on this point in the updated version of the paper.

---

### Official Review · Reviewer_AQjM · 2023-10-29

**Soundness:** 1 poor
**Presentation:** 2 fair
**Contribution:** 1 poor
**Rating:** 3
**Confidence:** 3

**Summary:**

This paper uses the QA templates from Liu et. al to prompt FLAN-T5 based multi-modal LLMs to generate NER tags. Experiments have been presented on the Twitter 2015, 2017 datasets which have multi-modal tweets with labels for persons, organizations and locations. Models used to study the QA based prompts include FLAN T5, BLIP2 and Instruct BLIP.  On both the Twitter 2015 and Twitter 2017 datasets, a vanilla FLAN t5 based prompt appears to do better than the use of multi-modal models which perhaps suggests that the hypothesis posed by the paper was found to be untrue. I was also unable to appreciate the aspect of low-resource NER tagging which the authors seem to suggest was one of their focus areas in the work. Perhaps the authors could clarify both points in the rebuttal.

**Strengths:**

None

**Weaknesses:**

Please see main summary.

**Questions:**

None

---

> ### Author Response · Authors · 2023-11-23
>
> Thank you for your review and your points regarding our paper. We appreciate the opportunity to address these concerns and provide further clarity.\
> Weaknesses:
> 1. The results do not point to the image modality concistently improving the results when considering overall micro-F1, and that’s part of the discussion and contribution of the paper.
> 2. If we are correct in saying that you ask us to elucidate the relevance of low-resource MNER, we have revised the paper to say more on this. Low-resource MNER is a broad context relevant to other tasks and also to unimodal NER.

---

### Official Review · Reviewer_xAYu · 2023-11-01

**Soundness:** 1 poor
**Presentation:** 2 fair
**Contribution:** 1 poor
**Rating:** 1
**Confidence:** 5

**Summary:**

The paper introduces a novel approach to Multimodal Named Entity Recognition (MNER) by framing it as an open-ended question-answering task, suitable for modern generative language models. The research offers insights into the complex interplay between model design, prompt construction, and training data characteristics affecting visual integration effectiveness. These findings inform future work in multimodal representation learning, generative modeling, and prompting at the intersection of text and image analysis.

**Strengths:**

This paper demonstrated low-resource MNER via prompting, with strong zero-shot performance. Analysis of our results and related work elucidates both progress and challenges for this emerging area.

**Weaknesses:**

1. The methodology section of this work primarily focuses on the design of instructions and lacks technical innovation, making it unsuitable for publication in ICLR.

2. I have several concerns regarding the statement "Results against state-of-the-art models for MNER are provided in Table 7" in Section 6: (1) Table 7 presents results from previous baselines in a fully supervised setting, but the experiments conducted earlier in this paper do not compare against these baselines. It is unclear how the conclusion "Results against state-of-the-art models" is derived. (2) The results of the method proposed in this paper are not included in Table 7. (3) To the best of my knowledge, Table 7 lacks citations and comparisons to papers in the MNER field from 2022-2023.

**Questions:**

How can we draw the conclusion with the statement "Results against state-of-the-art models for MNER are provided in Table 7"?

---

> ### Author Response · Authors · 2023-11-23
>
> We are sincerely grateful for your review and the opportunity to enhance our paper.\
> Weakness:\
> [1] We respect the reviewer’s opinion. At the same time, we would like to point to a lack of concrete arguments about what this weakness consists of. In support of our work, we would like to mention that papers studying prompting methods used in novel ways have been accepted to ICLR previously (Progressive Prompts: Continual Learning for Language Models, for example). We agree with the reviewer that the instruction part of our paper is not innovative.  However, we strongly argue that the multimodal NER setting is innovative. \
> [2] You correctly point out that we don’t have comparative results of low-resource MNER experiments. The reason for this is that we have not found other low-resource results for these datasets.  As a result, apples-to-apples comparisons indeed become challenging. Our assumption, on the other hand, was that our low-resource results should not have an unfair advantage over other researchers’ full-resource results.  (Most people would argue that it is the other way around.)  We have provided some more context and discussion for our results, emphasizing that our low-resource results are at a disdvantage to the full-resource results,  but it’s correct that this is a bit of an apples to oranges comparison.\
> Question:\
> [1] This was not worded well enough in the original version of the paper but has now been clarified.   We mean to say that we can look at our results in the context of fully fine-tuned results.

---

### Official Review · Reviewer_enH8 · 2023-11-01

**Soundness:** 2 fair
**Presentation:** 2 fair
**Contribution:** 3 good
**Rating:** 5
**Confidence:** 5

**Summary:**

In this paper, the author proposes an attempt to convert MNER tasks into "Instruction+QA" to solve the problem of labeled data absence, utilizing the excellent zero-shot learning ability of multimodal LLMs to address the low-resource MNER.  A series of question prompt templates are designed to effectively integrate visual cues in images. Experimentals demonstrate the effectiveness of the approach.

**Strengths:**

[1]Transforming multi-class named entity recognition (MNER) into an openended question-answering task is novel and effective in the experiments.

**Weaknesses:**

[1]The related works are not comprehensive enough. For example, it lacks some pre-trained models and multimodal NER models in the recent two years (2022-2023). It is unclear with the current presentation compared to previous works. For example, the below reference also considers the multimodal NER.
-Fei Zhao, Chunhui Li, Zhen Wu, Shangyu Xing, Xinyu Dai. Learning from Different text-image Pairs: A Relation-enhanced Graph Convolutional Network for Multimodal NER. ACM Multimedia 2022: 3983-3992.
-Lin Sun, Jiquan Wang, Kai Zhang, Yindu Su, Fangsheng Weng. RpBERT: A Text-image Relation Propagation-based BERT Model for Multimodal NER. AAAI 2021: 13860-13868.
-Lin Sun, Jiquan Wang, Yindu Su, Fangsheng Weng, Yuxuan Sun, Zengwei Zheng, Yuanyi Chen. RIVA: A Pre-trained Tweet Multimodal Model Based on Text-image Relation for Multimodal NER. COLING 2020: 1852-1862
[2]Based on the statistics on the datasets used, we can find that it is category unbalanced. Therefore, using the micro-f1 as the evaluation only may be not convincible.
[3]The author(s) need to comprehensively research the related datasets. It is not like the authors claimed that “Notably, these are currently two of the few openly accessible multimodal datasets for named entity recognition.”
-Dianbo Sui, Zhengkun Tian, Yubo Chen, Kang Liu, Jun Zhao.A Large-Scale Chinese Multimodal NER Dataset with Speech Clues. ACL/IJCNLP (1) 2021: 2807-2818
-Xigang Bao, Shouhui Wang, Pengnian Qi, Biao Qin.Wukong-CMNER: A Large-Scale Chinese Multimodal NER Dataset with Images Modality. DASFAA (3) 2023: 582-596
[4] The low-resource MNER ability of the proposed method mainly comes from the existing zero-shot learning of multimodal LLMs and a series of manually designed prompts, lacking further optimization for MNER.
[5]: Lacking the zero-shot and few-shot experimental results of existing MNER methods (such as a series of BERT based models and LLMs based models )  in evaluation.  Unable to accurately measure the advantages of the proposed method compared to other methods.
[6]All tmplates in the article are manually set, but there was no analysis of the experimental results with different tmplates. The impact of the different tmplates settings on LLMs for MNER should be noted.

**Questions:**

[1]It is not clear that the author(s) show the Zero-shot F1 in Figure 5 and Figure 6, which is not consistent with the previous claim. And the micro f1 scores by model with images is lower than that without images in Table 5 and Table 6? If there are some clerical errors?
[2]How to compute the c(A_i) is not clear.
[3]Table 4 is not referenced.

---

> ### Author Response · Authors · 2023-11-23
>
> We are grateful for your thoughtful feedback and the opportunity to refine our work. Forgive us for the late replies, we have tried to address the feedback pointwise below.\
> Weaknesses:\
> [1] It’s correct that not all works on MNER have been referenced.   This was due to limiting our references to the most relevant related work, as MNER is a broad task where the methods used differ widely. Two of the papers you mention (RpBERT and RIVA)  were in fact already referenced and included in Table 7 (now Table 6).  Further, we have included the last reference you mentioned in the revised version of the paper, as it is a highly relevant paper.  Thanks for the reference to this work.\
> [2] It’s a very good point that the micro-F1 score is not necessarily the best metric.  This was somewhat alleviated by including the entity-level F1 scores, which then alleviates this somewhat. Still, we have mentioned the evaluation metrics and other possible metrics possible in the revised paper. The further inclusion of precision and recall could have shown more nuance in the results, that’s correct, but also other metrics and so we refuge to mentioning this as caveats and suggest future work.\
> [3] Related literature (including the abovementioned papers) only handles the two mentioned Twitter datasets.  There is an additional dataset from Snapchat. However, it is not publicly available. You interestingly mention other available multimodal datasets. However, the first one is audio and text, not image and text, which we focus on, in addition to being in Chinese (while our focus is on English text). The last mentioned dataset is also in Chinese and was recently released in 2023, which is perhaps why we have overlooked it. Still, there seem to be very few datasets available, so while an interesting point, it’s not necessarily wrong. \
> [4] Indeed, there have not been added several optimizing steps in the process specifically for MNER, but on the other hand this is due to the method being interesting because it’s so versatile (simple prompting). Adapting to various contexts without needing specialized optimization makes it broadly applicable/usable/implementable.\
> [5] Such experimental results are lacking for the following reasons. Among the literature we reviewed, no other methods have done zero-shot ML on MNER as we have.  The reviewer correctly points out that we did not implement another method using BERT (for example) that could zero-shot. This is perhaps a good suggestion for future work.  It should be noted that there are many considerations to be made if we want to be able to zero-shot BERT (assuming with a custom classifier head) and also compare it to a larger causal LM. We assume that the low-resource case in our experiment, the “true few-shot setting” should not do better than full-resource case exemplified by the surveyed results in Table 7 (now 6), and we therefore thought it was a not perfect but acceptable comparison under constraints of limited time.\
> [6] This is a good point, and our reasoning was that it was not that interesting to show this (even though the standard deviations reflect it somewhat) due to there not being a  consistently outperformant prompt. Further,  the prompts themselves are not radically different, meaning it would be strange to say that one prompt is probably better than the other in this limited setting.\
> Questions:\
> [1] We are not entirely sure what you mean regarding Figures 5 and 6, as there are no Figures 5 and 6 in the paper, we assume that tables are what is meant, as the Zero-shot F1 scores are mentioned in Table 5 and Table 6 (now 4 and 5). Indeed, experiments with images sometimes have lower scores than those without. This is not a mistake. We state and reflect on this in the submitted paper, and this has been further clarified in the revision. \
> [2] This is a good point. We have clarified how to compute this in the revision. It’s the outputted logits from the model.\
> [3] Well pointed out, it should be fixed in the revision.

---

### Meta-Review · Area_Chair_8LV3 · 2023-12-18

**Metareview:**

The paper attempts to address low-resource Multimodal Named Entity Recognition (MNER) by ingeniously converting MNER tasks into an Instruction+QA framework and employing the zero-shot learning capabilities of multimodal large language models (LLMs). This is a commendable effort towards leveraging the capabilities of generative models for MNER, an area with relatively scarce datasets.

The reviewers identified several significant shortcomings of the work. They highlight that relevant recent works have been omitted, suggesting an inadequate literature review. Additionally, the results presented lack the necessary comparison to existing methods or robustness across different settings, impacting the paper's contribution to the field. There are also concerns that the visual modality doesn't uniformly enhance performance, questioning the multimodal approach's effectiveness.

**Justification For Why Not Higher Score:**

The reviewers identified several significant shortcomings of the work. They highlight that relevant recent works have been omitted, suggesting an inadequate literature review. Additionally, the results presented lack the necessary comparison to existing methods or robustness across different settings, impacting the paper's contribution to the field. There are also concerns that the visual modality doesn't uniformly enhance performance, questioning the multimodal approach's effectiveness.

**Justification For Why Not Lower Score:**

N/A

---

### Decision · Program_Chairs · 2024-01-16

Reject